# DNA-Based Molecular Engineering of the Cell Membrane

**DOI:** 10.3390/membranes12020111

**Published:** 2022-01-19

**Authors:** Xiaodong Li, Tiantian Wang, Yue Sun, Chang Li, Tianhuan Peng, Liping Qiu

**Affiliations:** 1Molecular Science and Biomedicine Laboratory (MBL), State Key Laboratory of Chemo/Biosensing and Chemometrics, College of Biology, College of Chemistry and Chemical Engineering, Aptamer Engineering Center of Hunan Province, Hunan University, Changsha 410082, China; xiao6150201076@163.com (X.L.); wtt10221998@163.com (T.W.); sunyue19970320@163.com (Y.S.); lc13618477890@163.com (C.L.); 2Key Laboratory of Birth Defect for Research and Prevention, Hunan Provincial Maternal and Child Health Care Hospital, Changsha 410005, China

**Keywords:** functional nucleic acids, DNA nanotechnology, cell membrane engineering

## Abstract

The cell membrane serves as a barrier and gatekeeper to regulate the cellular transportation of substances and information. It plays a significant role in protecting the cell from the extracellular environment, maintaining intracellular homeostasis, and regulating cellular function and behaviors. The capability to engineer the cell membrane with functional modules that enable dynamic monitoring and manipulating the cell-surface microenvironment would be critical for studying molecular mechanisms underlying various biological processes. To meet this goal, DNA, with intrinsic advantages of high versatility, programmability, and biocompatibility, has gained intense attention as a molecular tool for cell-surface engineering. The past three decades have witnessed the rapid advances of diverse nucleic acid materials, including functional nucleic acids (FNAs), dynamic DNA circuits, and exquisite DNA nanostructures. In this mini review, we have summarized the recent progress of DNA technology for cell membrane engineering, particularly focused on their applications for molecular sensing and imaging, precise cell identification, receptor activity regulation, and artificial membrane structures. Furthermore, we discussed the challenge and outlook on using nucleic acid materials in this specific research area.

## 1. Introduction

The cell membrane acts as a significant biological barrier and gatekeeper for maintaining a stable intracellular environment by regulating the exchange of substances with the extracellular matrix. The concentration and distribution of specific substances in the local surrounding of the cell membrane are essential for studying the molecular mechanism underlying many biological processes [1,2]. Meanwhile, the cells can sense, transduce, and respond to these substances mainly through membrane receptors, some of which have been regarded as potential targets for disease diagnosis and therapy [3]. Indeed, the capability to monitor the dynamic microenvironment surrounding the cell membrane, as well as manipulate specific membrane receptors, is critical to mediate various cellular functions and behaviors, such as proliferation, differentiation, apoptosis, and senescence [4]. To achieve this goal, engineering the cell membrane with functional tools is an essential strategy.

While genetic modification strategies have been widely used to engineer recombinant proteins onto the cell membrane, the complicated operation and potential alternation of the protein structure and expression hinders their practical application [5]. Meanwhile, intensive efforts have been made by chemical and biomedical communities to develop alternative strategies for cell-surface engineering [6]. DNA, taking advantage of easy synthesis, convenient modification, high programmability, and good biocompatibility, has become one of the most promising materials in this regard [7,8,9,10]. In addition to being the carrier of genetic information, DNA, with specific biological and chemical functions, including aptamers, DNAzyme, and molecular beacons, has been developed and attracted a broad interest for both biological and chemical research [11,12,13]. In the early 1980s, Thomas R. Cech discovered the self-splicing RNA that possessed intrinsic catalytic activity [14]. Then, functional nucleic acids with catalytic activity, mainly ribozymes, were obtained through in vitro selection [15,16]. In 1990, another representative FNA named aptamers, which could specifically bind with target cargos with a high affinity, were screened through an in vitro selection technology termed Systematic Evolution of Ligands by Exponential Enrichment (SELEX) [17,18]. Aptamers provided excellent molecular recognition ligands for targeting and regulating the activity of membrane receptors. Overall, the development of FNAs expanded the cognition of nucleic acids from genetic carriers to functional molecules, providing programmable tools for various biological applications [19,20].

Meanwhile, with the predictable base pairing rule, DNA has been used as the building block to develop various nanostructures with a precise size and shape, offering wide-spread applications in biomedical engineering [21,22,23,24]. Soon after the pioneered work of Nedrian C. Seeman, [25] many DNA nanostructures, including DNA origami, DNA framework, and dynamic DNA network, have been reported [26,27,28,29,30,31]. These DNA nanostructures showed multiple merits for cell membrane engineering [9], including the capability of controlling the structure and function on a nano-scale and high programmability and addressability for precise biological mimicry and manipulation [32,33]. In this review, we summarized recent progress in exploiting nucleic acids for cell membrane engineering (Figure 1). We focused on their applications in molecular sensing and imaging, cell identification, and activity regulation, and artificial membrane structures will be discussed. We also discussed the challenges and outlook of DNA technology in cell membrane engineering.

## 2. Membrane Functionalization Strategies

To develop a versatile DNA-based platform for monitoring and manipulating the cellular microenvironment, a primary requirement would be the efficient cell-surface engineering of functional DNA probes with limited interference on the cellular state [34]. Since the phospholipid bilayer is a major component of the cell membrane, the hydrophobic insertion would be an attractive approach for the membrane functionalization of amphiphilic DNA probes containing hydrophobic tags, such as cholesterol, tocopherol, aromatic derivatives, and other synthetic hydrophobic polymers (Figure 2) [35,36,37]. Liu et al. synthesized a series of diacyllipid-DNA conjugates and demonstrated their excellent capability for rapid and efficient membrane anchoring [38]. Qiu et al. then proved that DNA strands, as hydrophilic macromolecules with a negative charge, mainly located on the outer leaflet of the plasma membrane, were beneficial for analyzing the cellular interaction with the external environment. In addition to conjugation with hydrophobic tails, hydrophobic nucleotide analogues offered an alternative strategy for precise control over the site and the number of hydrophobic tags in the DNA sequence. For example, Wang et al. synthesized an artificial nucleotide analogue containing 3,5-bis(trifluoromethyl)benzene (F) that could be accurately incorporated into oligonucleotides through automatic solid-phase DNA synthesis [39,40]. Meanwhile, Abdullah et al. synthesized a hydrophobic nucleotide analogue with a ferrocene base (Fc-base) and demonstrated that DNA strands extended with an Fc-base could self-assemble into micelle structures, revealing great potential for cell-surface engineering [41,42]. To enhance the membrane anchoring stability of amphiphilic probes, several amphiphilic nanostructures have been utilized. For instance, Li et al. developed an innovative strategy for stable membrane anchoring of nucleic acids using a three-dimensional (3D) amphiphilic pyramidal DNA scaffold, enabling stable membrane modification of DNA probes [43]. Considering the poor selectivity of hydrophobic interaction, Jin et al. developed an alkaline phosphatase-dependent cell membrane inserting strategy [44]. The phosphorylated lipid conjugated oligonucleotide with poor hydrophobicity and displayed weak interactions with the lipidic membrane. After specific enzyme-catalyzed dephosphorylation, the phosphate group on the lipid end was eliminated. As such, the hydrophobicity of the lipid-conjugated oligonucleotide probe was enhanced, realizing the selective anchoring of oligonucleotides on the membrane of cells with high alkaline phosphatase expression.

Apart from hydrophobic insertion, covalent cross-linking, which utilized chemical active groups of surface proteins and glycans, offered another membrane modification strategy with high stability (Figure 2). As a representative, thiolated DNA probes could be engineered onto the cell surface through covalent conjugation with the cysteine and/or lysine residues on the membrane proteins [45,46]. Despite the high efficiency, these modification strategies might influence the proteins’ function by blocking their activity-related domains. As an alternative, carbohydrate metabolism could introduce specific chemical reactive groups onto glycoproteins through incubating cells with unnatural sugar [47]. For instance, Chandra et al. succeeded in decorating dibenzocyclooctyne (DBCO)-labeled DNA probes on the surface of cells pretreated with acetylated N-α-azidoacetylmannosamine via copper-free click chemistry [48]. In addition to modulating the cellular metabolism, Yousaf et al. introduced biorthogonal reactive groups onto the plasma membrane based on the delivery and fusion of functional liposomes [6]. After introducing ketone or oxyamine groups onto the cell surface, the chemoselective ligation of DNA probes could be realized through the formation of an oxime bond.

While these strategies allowed efficient membrane modification with DNA probes, they displayed low cell specificity. The capability to decorate functional DNAs on specific cells and even specific membrane proteins would offer new opportunities in cell studies. The genetic engineering strategy has been widely used to modify target membrane receptors with molecular tags through site-specific genetic fusion, which possibly led to the alternation of the natural protein structure [49]. Recognition ligands, such as antibodies, peptides, and aptamers, were used for targeting membrane proteins with molecular specificity (Figure 2). Aptamers, especially those screened through the cell-SELEX technology, enabled cell-type-specific recognition without prior knowledge of the cellular signature. Meanwhile, aptamers could be easily integrated with different DNA structures via hybridization or one-pot DNA synthesis, thus showing great flexibility for cell-surface engineering [50]. Many previous works have reported the cell membrane-targeting modification of oligonucleotide probes by using aptamers as the recognition ligand [51,52,53,54].

## 3. DNA-Based Molecular Engineering on Cell Membrane

### 3.1. FNAs for Cell Surface Sensing and Imaging

The extracellular microenvironment refers to a dynamic local surrounding, which plays a significant role in mediating the cellular interaction with external environments. Real-time monitoring of this microenvironment would provide valuable information for studying the molecular mechanism underlying various biological processes, such as cell signaling, communication, metabolism, and differentiation. FNAs, taking advantage of flexible design and convenient modification, showed great potential for monitoring the variation of specific molecules or conditions in the extracellular microenvironment.

For example, Liu et al. engineered the cell surface with a pH-responsive DNA machine and demonstrated its good performance for dynamically sensing the pH variation in the extracellular environment with high sensitivity [55]. By using a cholesterol-labeled DNA tweezer as the scaffold, Yao et al. engineered a pH-responsive nano-construct onto the cell surface [56]. Based on the pH-dependent configuration switch of DNA i-motif, this nano-construct allowed monitoring of the pH fluctuation with high spatial and temporal resolution (Figure 3A,B). In addition to the pH condition, metal ions served as important modulators that participated in many physiological processes, including substance transportation, energy conversion, and metabolic regulation. DNAzymes, which used specific metal ions as the co-factor in mediating the enzymatic activity, provided excellent molecular tools for sensing and imaging of target ions. Representatively, a Mg^2+^-specific DNAzyme was conjugated with a diacyllipid tail and anchored onto the cell surface for dynamically monitoring the cellular efflux process of the target ion [57]. Meanwhile, based on the K^+^-dependent quadruplex structure transformation, Xiong et al. developed a K^+^ responsive fluorescent sensor that could monitor the fluctuation of K^+^ in the extracellular microenvironment [58]. Additionally, aptamer-based sensing probes have been built on the cell membrane. For example, Tokunaga and co-workers developed a membrane-anchored fluorescent aptamer sensor for monitoring the rapid cellular secretion of adenine, an important neurotransmitter, from brain astrocytes under external stimulation [59]. In addition, other cell-secreted signaling molecules, such as sulfur dioxide derivatives and nitric oxide, were probed by membrane-anchored DNA sensors. For ratiometric imaging of signaling molecules, Feng and co-workers developed a modular strategy by engineering DNA motifs and synthetic cofactors [60]. As shown in Figure 3C,D, the dynamic cellular extrusion process of endogenous signaling molecules could be monitored based on fluorescence resonance energy transfer (FRET).

In addition to monitoring specific substances surrounding the cell surface, the dynamics of membrane composition would be another important indicator of the cellular status. You et al. used amphiphilic DNA probes as the building block and developed a smart strategy for monitoring the transient membrane encounter events through toehold-mediated DNA strand displacement reaction (Figure 3E) [61]. By conjugating DNA probes with different hydrophobic tails, including diacyllipid, cholesterol, and tocopherol, this strategy showed potential for studying the interaction dynamics of different membrane domains. Soon after, a similar strategy was reported for monitoring the interaction of membrane proteins by exploiting aptamers as the targeting ligand [62].

### 3.2. DNA-Based Intercellular Communication

In multicellular organisms, cell–cell communication was mainly mediated by membrane receptors. The development of artificial receptors to monitor and control intercellular reactions would be highly desired. Cell membrane-anchored DNA platforms, with high programmability and flexibility, showed great promise in mimicry and manipulation of cellular interaction.

Based on DNA hybridization with predictable thermodynamics, Zhao et al. reported DNA tension probes for quantifying the tensile forces at the junction between cells. As shown in Figure 4A,B, the tension probe was inset into the cellular junction through both the hydrophobic interaction and the ligand–receptor interaction [63]. Upon sensing the intercellular tension force, the DNA probe would undergo a conformation switch to separate the fluorophore pair, thus allowing dynamic detection of the intercellular force via FRET. Moreover, the capability to modulate the intercellular reaction would offer us a straightforward strategy for analyzing related biological processes. Xiong et al. mimicked the natural cell–cell adhesion by modifying the cell surface with specific aptamers (Figure 4C,D) [52]. Based on the aptamer-mediated molecular recognition, the effector T cells could specifically bind with target cancer cells, resulting in enhanced cancer-killing efficacy. To achieve intelligent control over the cellular interaction, DNA-based Boolean logic operation was utilized [48]. Li et al. engineered an amphiphilic DNA tetrahedral construct on the cell surface, which could undergo a conformation switch and assemble functional modules for sensing the cell’s adaptive response to the external environment [64]. As such, this membrane-anchored DNA construct allowed regulation of the cellular interaction coordinated with the cellular activity.

Another application for membrane anchored-DNA probes was to manipulate the morphology of cells with spatial resolution. The cell growth could be controlled via hybridization of the membrane-anchored DNA with the complementary DNA on the substrate interface [65]. Meanwhile, a 3D cell microsphere could be constructed by stepwise and programmable cell assembling, which displayed great promise for the construction of 3D artificial tissues [66]. In addition to single-stranded DNA probes, DNA nanostructures provided versatile tools for controllable cell assembly [47]. For example, the assembly between stem cells and niche cells was mediated through the mechanical and morphogen signaling from niche cells, providing a versatile strategy for the generation of stem cells in vitro [64].

### 3.3. DNA-Based Receptor Monitoring and Regulating

Protein receptors expressed on the cell surface played a pivotal role in regulating cellular function and behavior. Abnormal expression and dysfunction of surface receptors were closely related with the occurrence of many diseases. As previously reported, over 50% of commercially available therapeutic agents targeted cell surface receptors. DNA nanotechnology, taking advantage of high programmability, high addressability, and good biocompatibility, offered promising strategies for the detection and regulation of surface receptors. Particularly, aptamers, selected by the cell-SELEX technology, offered a panel of molecular tools for specific recognition of surface receptors [67,68,69].

Widespread application of aptamers for targeted cell imaging has been realized in the past two decades. [70,71] Their application for super-resolution imaging was emerging as an interesting topic. To image the membrane receptor at the single-molecule level, Delcanale et al. used aptamers as the molecular recognition ligand and developed DNA-based point accumulation for imaging in nanoscale topography (DNA-PAINT) (Figure 5A) [51]. The stochastic and transient binding against target receptors could be realized through regulating the binding affinity of aptamers, providing twinkling fluorescence signals for super-resolution imaging of the cell membrane. Recently, this strategy was further expanded to visualize the spatial proximity of membrane receptors using a split-docking site configuration [72].

Cell identification with high accuracy was fundamental in the perspective of precise medicine. Conversely, due to their high heterogeneity, it was rather challenging to identify specific cells with single makers. Exploiting multiple membrane biomarkers served as one of the promising strategies for improving the accuracy of cell identification. Meanwhile, to achieve intelligent cellular identification, DNA circuits have been incorporated to operate smart computation of multiple biomarkers on the cell membrane. As one of the typical examples, Rudchenko et al. reported an autonomous molecular machine based on DNA strand replacement reactions for precise cellular identification, which allowed distinguishing the subpopulation of lymphocytes from human blood cells [73]. To improve the efficiency and accuracy of DNA computation, Peng et al. designed a 3D DNA nanomachine that integrated all logic gate elements on one DNA construct (Figure 5B) [53]. A decision signal output could be generated only upon recognizing target cancer cells synchronously expressed with two membrane markers.

In addition to detecting the expression level of surface receptors, Li et al. developed an aptamer-based fluorescence probe for dynamically monitoring the dimerization of Met receptors [74]. The same group then developed a bispecific aptamer for mediating the heterodimerization between Met and TfR receptors, which could inhibit the formation of a Met homodimer mainly via a steric-hindrance effect, thus regulating downstream signaling pathways (Figure 5C) [62]. Meanwhile, the cellular morphology could be regulated by membrane-anchored DNA nanostructures, which then affect the motion and function of cells. Furthermore, the addressability of DNA origami enabled precise control over the spatial organization of membrane ligands. In 2014, Shaw et al. used hollow tube-like DNA origami to develop a “nanocaliper” and proved its feasibility for regulating the activation process of EphA2 through manipulating the spatial distribution of ephrin-A5 on the cell membrane [75].

### 3.4. DNA-Based Biomimetic Membrane Constructs

Protein channels on the cell membrane are one of the most important organelles for substance exchange with the external matrix and play significant roles in cell–cell communication. The programmability and site addressability of DNA nanostructures make them ideal materials for mimicking the structure and function of membrane channels. In 2016, Howorka et al. reported a DNA nanopore with hollow barrel structures for regulating the transportation of small molecules across the plasma membrane (Figure 6A) [76]. Specifically, the modification of hydrophobic cholesterol at the outside of the constructed DNA origami enabled the insertion of nanopores into the lipidic membrane, and the “closed” and “open” state of nanopores could be regulated with “lock” and “key” strands. They demonstrated that these artificial ion channels were able to control the transportation of small organic molecules with high selectivity. Soon after, Diederichs et al. constructed a synthetic DNA nanopore with a larger size, which enabled transportation of folded proteins [77].

In addition to constructing artificial nanopores for substance transporting, cell membrane-anchored DNA nanostructures could be designed as bio-mimic receptors to regulate the cell signaling process. In 2020, Peng et al. designed a DNA-based artificial signal transduction system with giant vesicles derived from living cells (Figure 6B) [78]. Adenosine triphosphate (ATP) responsive artificial nanopores were developed and anchored onto the vesicles through hydrophobic interactions. Upon exposure to ATP, the locker strand of nanopore would be released, leading to the import of metal ions, which then initiated a DNA circuit reaction to mimic the signaling cascades.

Meanwhile, specific DNA nanostructures anchored on the cell membrane could also display enzyme-like catalytic activity. For example, Göpfrich et al. reported that membrane-inserted dsDNA with porphyrin modification could locally induce the formation of hydrophilic pores [79]. They continued to construct a hollow barrel structure composed of four double helices. After being inserted into the cell membrane, this nanostructure displayed scramblase activity for promoting transportation of lipids between membrane leaflets, achieving a three-fold higher catalytic rate than protein enzymes [80].

## 4. Conclusions and Perspectives

DNA, taking advantage of high programmability, addressability, and biocompatibility, offered promising materials for bioengineering. In the past decade, enthusiasm over the utility of DNA constructs for cell membrane engineering with designated structures and functions was increasingly coming into view, offering new opportunities for various biological studies. While considerable progress has been achieved, several technical challenges still remain. Firstly, the stability of DNA structures on the membrane should be enhanced. Meanwhile, effective strategies for precise synthesis and purification of DNA structures on a large scale were highly desired. Moreover, more efforts were needed to push forward the application scope of membrane-anchored DNA platforms from in vitro to in vivo. Last but not least, deep and broad collaboration was urgently needed among multiple disciplines, such as advanced materials, molecular biology, and chemistry, to promote this innovative field.

## Figures and Tables

**Figure 1 membranes-12-00111-f001:**
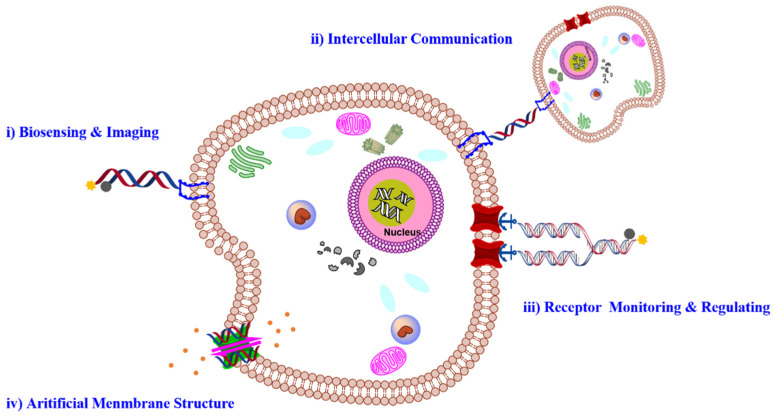
Schematic illustration of different applications using nucleic acid materials on a cell membrane, mainly including molecular sensing and imaging, intercellular communication, receptor monitoring and regulating, artificial membrane structure.

**Figure 2 membranes-12-00111-f002:**
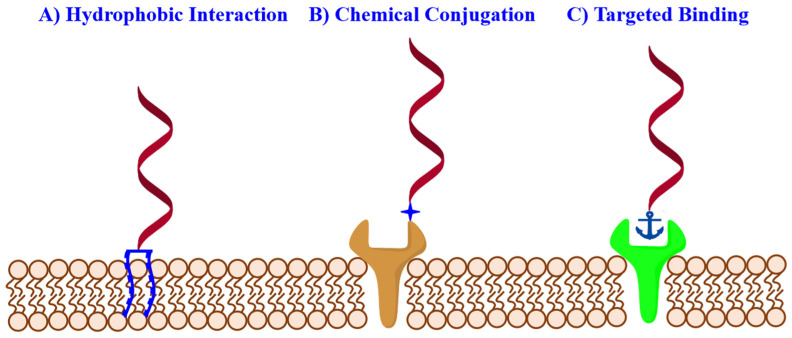
Illustration of different membrane functionalization strategies for cell-surface engineering. (**A**) Hydrophobic Interaction: hydrophobic insertion with amphiphilic DNA conjugates, the hydrophobic tags include cholesterol, tocopherol, aromatic derivatives, and other synthetic hydrophobic polymers. (**B**) Chemical Conjugation: the modified DNA can anchor to the cell membrane through covalent cross-linking with chemical active functional groups on membrane proteins and glycans. (**C**) Target Binding: functional DNAs can be decorated on specific cells and specific membrane proteins using genetic engineering strategy or recognition ligands, such as antibodies, peptides, and aptamers.

**Figure 3 membranes-12-00111-f003:**
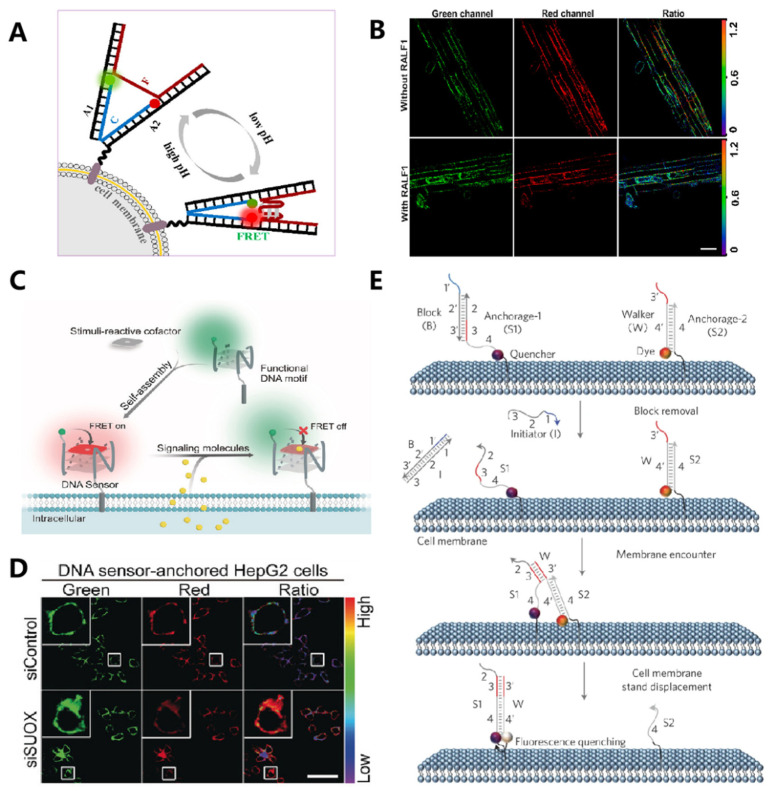
Application of FNAs for cell surface sensing, imaging and monitoring. (**A**,**B**) Construction of the pH-sensitive DNA tweezer for real-time monitoring of extracellular and apoplastic pH. Reprinted with permission from Zeng et al. [49], Copyright 2018, with permission from Elsevier. (**C**,**D**) Imagin of the released sulfur dioxide derivatives and nitric oxide from an intracellular environment through DNA motifs and synthetic cofactors-based DNA sensors. Reprinted with permission from Feng et al. [53], Copyright 2019, with permission from Elsevier. (**E**) DNA probes can be anchored on the live cell membrane to monitor dynamic and transient molecular encounters. Reprinted with permission from You et al. [54], Copyright 2017, with permission from Elsevier.

**Figure 4 membranes-12-00111-f004:**
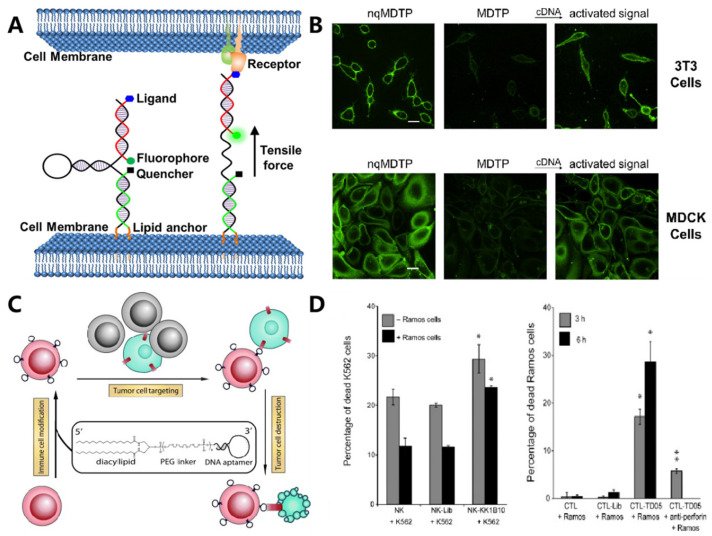
DNA-based probes mediated non-destructive modification on the cell surface and cell–cell interaction. (**A**,**B**) Lipid-modified DNA probes to visualize tensile forces at cell junctions. The intercellular tension force can open the DNA probes and large increase in the fluorescence intensity. Reprinted with permission from Zhao et al. [56], Copyright 2017, with permission from Elsevier. (**C**,**D**) Immune cells were modified with lipo-aptamer probes to enhance the cancer cell targeting and killing efficiency (The single asterisk indicates a significant difference between aptamer-modified and unmodified or Lib-modified groups determined by the one-tailed *t*-test at * *p* < 0.01, ** *p* < 0.001. The double asterisks indicate a significant difference between aptamer-modified and anti-Perforin treated groups determined by the one-tailed *t*-test at * *p* < 0.01). Reprinted with permission from Xiong et al. [52], Copyright 2013, with permission from Elsevier.

**Figure 5 membranes-12-00111-f005:**
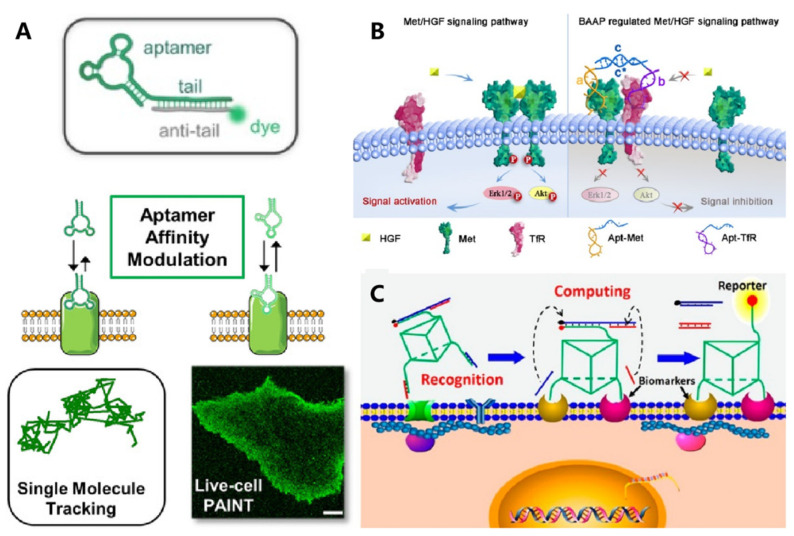
DNA-based molecule probes to monitor and regulate cell surface receptors. (**A**) Aptamer probes were used for single-molecule PAINT imaging to track and localize the membrane receptors on living cancer cells. Reprinted with permission from Delcanale et al. [44], Copyright 2020, with permission from Elsevier. (**B**) Bispecific aptamer induced protein dimerization to specifically regulate Met receptor function and downstream signaling pathways. Reprinted with permission from Wang et al. [55], Copyright 2019, with permission from Elsevier. (**C**) 3D DNA-logic gate nanomachine was constructed to recognize and compute the protein receptors on target cell surfaces. Reprinted with permission from Peng et al. [46], Copyright 2018, with permission from Elsevier.

**Figure 6 membranes-12-00111-f006:**
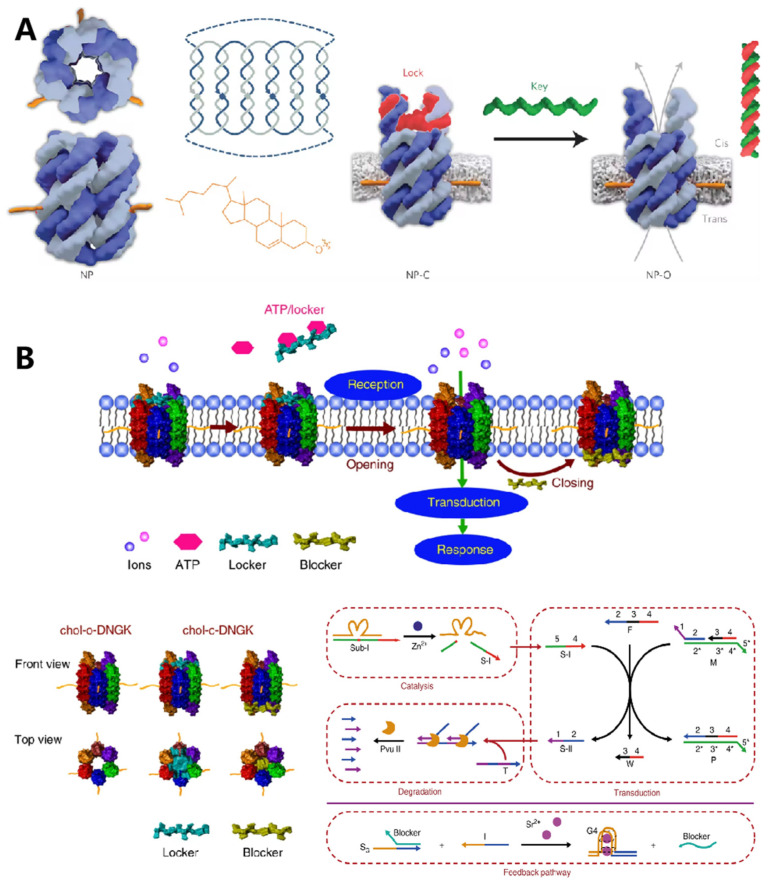
DNA nanostructures in constructing biomimetic membrane. (**A**) The typical DNA nanopore with a nanomechanical and sequence-specific gate to regulate the transportation of small molecules across the cell membrane. Reprinted with permission from Burns et al. [69], Copyright 2016, with permission from Elsevier. (**B**) DNA-based artificial signal transduction system with a cell-mimicking giant membrane vesicle. Reprinted with permission from Peng et al. [71], Copyright 2020, with permission from Elsevier.

## Data Availability

Data are contained within the article.

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
