# Peer review of "DNA-Based Molecular Engineering of the Cell Membrane"

_membranes, 2022, doi:10.3390/membranes12020111_

Round 1
Reviewer 1 Report
DNA continues to amaze us with its functions and applications. DNA not only carries genetic information, but is also used as building blocks for various synthetic functional materials. The present manuscript summarizes the progress of DNA technology for the cell membrane engineering, the application of this materials for cell surface sensing and imaging, DNA based cell communication, receptor activity regulation and artificial membrane structures.
In my opinion, this work meets the highest standard and I recommend that the manuscript be accepted.
I have some suggestions for adjustments: On page 3 line 96 the line “… For instance, Li et al. developed an innovative strategy for stable membrane anchoring of nucleic acids using a …” it is not quite clear which reference it is related to.
The abbreviation of Analytical Chemistry is written wrong in several references.
Author Response
We greatly appreciate the reviewer’s comments and suggestion on this work. In the revised manuscript, we added the reference mentioned by the reviewer and corrected the abbreviation of Analytical Chemistry in corresponding references.
Reviewer 2 Report
In this review, the authors cover the applications of DNA nanotechnology on cell membrane, which is a significant subject. The authors describe the impact of cell membrane microenvironment on biological processes, and point out the importance to monitor the surrounding environment and manipulate specific membrane receptors to mediate cellular functions and behaviors. Then, the authors introduce the unique properties of DNA, the versatile functions, and the programmable design of DNA nanostructures, which shows multiple merits of DNA nanostructures for cell membrane engineering. Starting from the functionalization of DNA on cell membrane, the authors guided the reader along with the applications of DNA-based molecular engineering on cell sensing and imaging, intercellular communication, receptor monitoring and regulation, and biomimetic membrane constructs. The conclusions and perspectives section presents the challenges and opportunities of DNA-based molecular engineering in this field. The review is well organized and written, and will benefit the development of DNA nanotechnology in cell biology field. I recommend this review for publication in Membranes after some minor revisions have been made.
- The following reference is suggested to be cited in the paragraph about biological and chemical research on page 2, ref. 11-12.
Wang, Fuan, Xiaoqing Liu, and Itamar Willner. "DNA switches: from principles to applications." Angewandte Chemie International Edition4 (2015): 1098-1129. - The following reference is suggested to be cited at the end of the first paragraph about functional nucleic acids on page 2, ref. 18.
Vázquez-González, Margarita, and Itamar Willner. "Aptamer-Functionalized Hybrid Nanostructures for Sensing, Drug Delivery, Catalysis and Mechanical Applications." International Journal of Molecular Sciences4 (2021): 1803. - The following reference is suggested to be mentioned in the paragraph about DNA nanostructures on page 2, ref. 19-21.
Liao, Wei‐Ching, and Itamar Willner. "Synthesis and Applications of Stimuli‐Responsive DNA‐Based Nano‐and Micro‐Sized Capsules." Advanced Functional Materials41 (2017): 1702732. - The following references are suggested to be mentioned in the paragraph about dynamic DNA network on page 2, ref. 23-26.
Liu, Xiaoqing, Chun-Hua Lu, and Itamar Willner. "Switchable reconfiguration of nucleic acid nanostructures by stimuli-responsive DNA machines." Accounts of chemical research6 (2014): 1673-1680.
Yue, Liang, et al. "Nucleic Acid Based Constitutional Dynamic Networks: From Basic Principles to Applications." Journal of the American Chemical Society142.52 (2020): 21577-21594. - The following reference is suggested to be mentioned in the paragraph about biological mimicry and manipulation on page 2, ref.27.
Wang, Jianbang, et al. "Active generation of nanoholes in DNA origami scaffolds for programmed catalysis in nanocavities." Nature communications1 (2019): 1-10.
Author Response
We greatly appreciate the reviewer’s comments and suggestion on this work. We have cited all the references suggested by the reviewer in the revised manuscript.